# Monkeypox Vaccine Acceptance among Ghanaians: A Call for Action

**DOI:** 10.3390/vaccines11020240

**Published:** 2023-01-21

**Authors:** Ramy Mohamed Ghazy, Saja Yazbek, Assem Gebreal, Mai Hussein, Sylvia Agyeman Addai, Ernestina Mensah, Michael Sarfo, Agyapong Kofi, Tareq AL-Ahdal, Gilbert Eshun

**Affiliations:** 1Tropical Health Department, High Institute of Public Health, Alexandria University, Alexandria 21561, Egypt; 2Faculty of Public Health, Lebanese University, Beirut 6573, Lebanon; 3Alexandria Faculty of Medicine, Alexandria University, Alexandria 21568, Egypt; 4Clinical Research Administration, Alexandria Directorate of Health Affairs, Alexandria 21554, Egypt; 5Ministry of Health and Population, Cairo 71529, Egypt; 6Department of Science & Department of Educational Administration and Management, University of Education, Winneba CE-119-9961, Ghana; 7Business School, Datalink University, Tema GN-03-266605, Ghana; 8Human Resource & Marketing Department, Central University, Miotso-Prampram GT-001-5571, Ghana; 9Department of Midwifery, Seventh Day Adventist Nursing and Midwifery Training College, Agona-Asamang P.O. Box GN 37, Ghana; 10School of Human and Health Sciences, University of Huddersfield, Huddersfield HD1 3DH, UK; 11Manhyia Government Hospital, Kumasi AK-039-5028, Ghana; 12Institute of Global Health (HIGH), Heidelberg University, Neuenheimer Feld 130/3, 69120 Heidelberg, Germany; 13Seventh Day Adventist Hospital, Agona-Asamang AZ-0581-8540, Ghana

**Keywords:** human monkeypox, monkeypox vaccine, psychological antecedents, 5C scale, Ghana, vaccine confidence, vaccine hesitancy, Sub-Saharan Africa

## Abstract

**Background**: Ghana ranked 31st worldwide and 3rd in Africa in the number of confirmed cases worldwide. We aimed to assess the intention to receive the monkeypox (MPOX) vaccine and its associated psychological antecedents among the Ghanaian population. **Methods**: A cross-sectional online survey was conducted in Ghana from November to December 2022. Snowball sampling was used to recruit participants via social media platforms, such as WhatsApp, LinkedIn, Telegram, and Facebook. The validated 5C scale was used to assess five psychological factors that influence vaccination behavior and intent: confidence, complacency, constraints, calculation, and collective responsibility. **Results**: The study drew 605 participants; their mean age was 30.0 ± 6.8; 68.1% were single; 60.8 % were males, and 51.9% were living in Greater Accra (The capital and largest city of Ghana). About 53.9% of the studied Ghanaian population did not intend to receive the MPOX vaccination. Vaccine acceptance among non-healthcare workers (non-HCWs) was significantly lower than among HCWs (41.7 vs. 55.3, *p* < 0.001). The determinants of vaccine acceptance were male gender (AOR = 1.48, 95% CI, 1.00–2.18, *p* = 0.049), urban residence (AOR = 0.63, 95% CI, 0.41–0.96, *p* = 0.033), refusal of coronavirus 2019 vaccine (AOR = 0.29, 95% CI, 0.16–0.52, *p* < 0.001), confidence in vaccination ((AOR = 2.45, 95% CI, 1.93–3.15, and *p* < 0.001), and collective responsibility (AOR = 1.34, 95% CI, 1.02–1.75, *p* = 0.034)). **Conclusions**: The participants in this study did not show high levels of intention to accept the MPOX vaccination. Consequently, tailoring the efforts aiming to promote MPOX vaccination is needed especially among non-HCWs through increasing their confidence in vaccine effectiveness and safety and promoting the importance of self-vaccination to protect others.

## 1. Introduction

Monkeypox (MPOX) is a re-emerging uncommon zoonotic infectious disease caused by the monkeypox virus (MPOXV), a large double-stranded DNA virus [1]. MPOXV belongs to the *Orthopoxvirus* genus and *Poxviridae* family, and it is more stable than RNA viruses in recognizing and fixing mutations [2]. The MPOXV was discovered in 1958, following two outbreaks of a pox-like disease in research-held monkeys [3]. Humans were first exposed to the disease in 1970 when a youngster in the Democratic Republic of the Congo was suspected of carrying smallpox. In 2003, the United States of America (USA) reported the first MPOX outbreak outside of Africa [3,4]. Following this outbreak, a number of reported cases of monkeypox and limited-spread outbreaks have been linked to travel or imported animals from the endemic regions [5].

The growing global MPOX outbreak was declared a Public Health Emergency of International Concern by the World Health Organization (WHO) Director-General on 23 July 2022 [6]. Since 1 January 2022, 110 Member States from all six WHO regions have reported cases of monkeypox to the WHO. As of 10 January 2023, at 17 h CET, the WHO has received reports of 84,415 laboratory-confirmed cases and 1348 suspected cases, including 76 deaths from 110 countries. Cases and long-term transmission chains have been documented for the first time in countries with no direct or immediate epidemiological linkages to West or Central Africa. When compared to week 52 (26–31 December) (n = 518 cases), the number of weekly reported new cases declined by 66.6% in week 1 (1–7 January) (n = 330 cases). However, the number of cases reported did not follow a fixed pattern across different weeks. The majority of cases recorded in the last four weeks were from the Americas Region (93.3%) and the European Region (5.3%) [7]. Ghana is considered one of the monkeypox-endemic countries representing the first source of the outbreak outside Africa in 2003 [8]. As of 10 January, Ghana ranked 31st worldwide and 3rd in Africa after Nigeria and the Democratic Republic of the Congo in the number of confirmed cases worldwide with 116 confirmed cases and four deaths [7].

The general population, including healthcare workers (HCWs) at all levels, should implement appropriate prevention and control measures against MPOX [9]. The most effective method to manage and contain an outbreak is believed to be through the judicious use of vaccines and preventative measures preventing the spread of the infectious disease from human to human and limiting the zoonotic disease’s transmission [1,10]. According to the literature, smallpox vaccination covers approximately 85% protection against MPOXV infection via cross-protection as members of the same poxviridae family [11,12]. Global attempts to create effective and safe vaccinations are ongoing, with approved vaccines already being used by high-risk groups in countries such as the USA, Canada, and the United Kingdom, among others. However, like with coronavirus disease (COVID-19), in order to achieve adequate vaccination coverage, people’s vaccine hesitancy (VH) to accept the vaccine is an important factor that vaccination programs should address. Fear of vaccine adverse effects, concerns regarding vaccine safety, efficacy, and effectiveness, lack of information, short duration of clinical trials, and social trust were the primary factors identified as influencing population attitudes regarding vaccination [13,14,15,16,17].

VH is considered an urgent threat at the individual and community levels eroding outbreak control and prevention efforts. As a result, communities become potentially susceptible to outbreaks of infections. In 2019, the WHO identified VH as one of the top ten threats to global health [18]. According to the Strategic Advisory Group of Experts (SAGEs) on Immunization, VH simply refers to a person’s delay or refusal to be vaccinated despite the availability of a safe and effective vaccine [19]. VH varies with time, place, and vaccines and is impacted by several variables, including complacency, convenience, and confidence [20]. Unlike available tools that only implement the 3C model (confidence, complacency, and constraints), the 5C scale expands the measures’ effectiveness because it evaluates the five psychological antecedents that influence an individual’s vaccination decision as confidence, complacency, constraints, calculation, and collective responsibility [21]. The 5C has been extensively used before to assess VH toward influenzas [22], COVID-19 [23], and MPOX [24].

The attitude of the general population including HCWs toward MPOX vaccinations in Ghana has not yet been the subject of any studies. We believe that studying the psychological antecedents of the general population including HCWs may give a more in-depth understanding of the individual mental representations, attitudes, and behaviors. In this study, we aimed to assess the intention of the Ghanaian population including HCWs to receive the MPOX vaccine. Moreover, we assessed the role of the psychological antecedents on the intention of the studied population to get vaccinated using the 5C scale.

## 2. Materials and Methods

### 2.1. Study Design and Study Setting

Using the snowball sampling technique, a cross-sectional online survey was undertaken in Ghana from 27 November to 6 December 2022 using Google forms. After presenting the study’s aims to participants, they were requested to take part through various social media platforms (i.e., WhatsApp, Facebook, Telegram, and LinkedIn). Those who expressed an interest in taking part in the study were invited to complete an online form of a self-administered questionnaire (Appendix A).

### 2.2. Study Participants and Sample Size

We recruited only the Ghanaian population including HCWs who were 18 years and above and had access to the Internet and smartphone or computer devices. As there were no studies published in Ghana to assess the attitude of the population including HCWs toward the MPOX vaccine, we supposed that 50.0% of the participants were willing to receive the MPOX vaccine. The sample size was calculated according to the following formula: [25]
Sample size n = [DEFF × Np (1 − p)]/[(d^2^/Z^2^1 − α/^2^ × (N − 1) + p × (1 − p)]

n = the minimum number of respondents required; Z^2^ = (1.96)2 relative to the 95% confidence interval (CI); P = (50%) the prevalence rate estimated in the previous study; e = the required accuracy (5%), design effect of 1, and no-response rate of 30%. The minimum sample size n for this study was 550 participants.

### 2.3. Studied Variables

The dependent variable was the intention to receive the MPOX vaccine (yes, no), while the independent factors included sociodemographic data, medical history, COVID-19 history, history of MPOX infection, and the 5 domains of the 5-scale (constraint, complacency, calculation, confidence, and collective responsibility).

### 2.4. Data Collection Tool

A self-administered questionnaire, in the English language, with three main sections that were tested and validated beforehand, was used in this study [21]. In the first section, the sociodemographic details of the study participants were documented. They included age, gender, nationality, the country where they are living now, marital status, living area and region, financial status, education level, chronic illness, and occupation (physician/physician assistant, nurse/midwife, pharmacist/pharmacy technician, laboratory technician, public health personal, not a healthcare worker). In section two, questions were asked if the person had received the COVID-19 vaccine, if they had been exposed to MPOX, if they knew somebody who had died from it, and their own intention to receive the MPOX vaccination. The 5C scale was covered in the third section, which evaluated the psychological antecedents of the Ghanaian population. This section consists of 15 questions categorized into 5 subscale areas, with responses organized into 7 Likert scale categories and optional hierarchy orders. “Extremely agree/somewhat agree/agree/neither agree nor disagree/disagree/somewhat disagree/disagree/extremely disagree”. The 5C subscale questions are as follows: confidence (Q1–Q3), complacency (Q4–Q6), constraints (Q7–Q9), calculation (Q10–Q12), and collective responsibility (Q13–Q15). We piloted the questionnaire before circulation, and it took 5–7 min to complete the questionnaire. The internal consistency of the 5 domains of the 5C scale was above 0.8 (good reliability). We followed the Strengthening the Reporting of Observational Studies in Epidemiology (STROBE). Appendix A [26].

### 2.5. Operational Definitions

Confidence refers to trust in the vaccine’s dependability and effectiveness, along with trust in the healthcare system and HCWs. A lack of confidence and mistrust leads to lower vaccine uptake, declined trust in the healthcare system, and growing recognition of misinformation [20]. The term “constraint” focuses on the structural and psychological barriers that may prevent people from getting vaccinated even if they intend to. Such barriers include access, time, self-efficacy, empowerment, and a lack of behavioral control [21]. Complacency is when the person perceives that the risks of vaccine-preventable diseases are low and vaccination is not deemed a necessary preventive action [27]. Calculation implies that people attempt to acquire information in order to weigh the risks of infections versus vaccination and make a well-informed decision. It is contended that calculation is a sign of risk aversion and may have a negative impact on vaccination behavior [21]. Collective responsibility is identified as “the willingness to protect others by immunizing oneself through herd immunity.” In other phrase, it applies to people who vaccinate themselves in order to protect others and better understand the role of herd immunity in restricting transmission [13].

### 2.6. Statistical Analysis

The R 4.2.1 program was used to manage and analyze the data (R Foundation for Statistical Computing, Vienna, Austria). The mean standard deviation (SD) was used to display numerical data, while nominal and categorical variables were reported as a percentage (%). The association between the qualitative factors was assessed using a Chi-square test, and responses were classified as Yes or No depending on obtaining COVID-19 booster doses. To examine the difference between the means of two independent groups, an independent *t*-test was used. A binary logistic regression analysis was performed to determine the odds ratios of the important predictors, and a 95% confidence interval (OR, 95% CI) was given. The dependent variable was virtual acceptance of MPOX vaccination, which was characterized by the following questions: Will you receive the MPOX vaccine “(Yes/No)”. A statistically significant *p*-value of 0.05 was used.

### 2.7. Ethics

This study was approved by the ethical committee of the Faculty of Medicine IRB (#0305708). The objectives of the study were indicated in the questionnaire’s introduction, and each participant was given the option to agree or decline to take part in it. All participant information was handled privately and anonymously.

## 3. Results

### 3.1. Participants’ Demographics

The study drew 605 participants; their mean age was 30.0 ± 6.8 ranging between 18.0 and 67.0 years; 60.8% were males; 74.8% were living in urban areas; over three fifths (68.1%) were single; 47.9% had a Bachelor’s degree; 54% were of low income; about half of them (51.9%) lived in Greater Accra; 67.4% were not HCWs; 4.5% had chronic diseases; 66.0% were fully vaccinated against COVID-19; 2.6% knew someone who passed due to MPOX, and 8.9% had contracted MPOX. The mean score confidence score was 5.7 ± 1.1; the constraints score was 4.3 ± 1.6; the complacency score was 3.9 ± 1.8; the calculation score was 5.8 ± 1.1, and the collective responsibility score was 5.3 ± 1.0 (Table 1).

### 3.2. Intention to Receive MPOX Vaccine

About 53.9% of the studied Ghanaian population did not intend to receive the MPOX vaccination, while 46.1% had the intention to receive the MPOX vaccine.

### 3.3. Association between Different Population Criteria and Intention to Receive MPOX Vaccine

Among the studied characteristics, older ages were significantly associated with higher vaccine acceptance in comparison to those who rejected vaccination (30.8 ± 7.1 vs. 29.4 ± 6.5, *p* = 0.013). Acceptance of the MPOX vaccine was more among males compared to females (60/3% vs. 49.7%), *p* = 0.013. MPOX vaccine acceptance among non-HCWs was lower than among HCWs (41.7% vs. 55.3%), and this difference was statically significant, *p* < 0.001. There were no statistically significant differences in MPOX vaccine acceptance across marital status, educational level, having chronic disease, income, region, or knowing someone who passed due to MPOX infection.

Fully vaccinated against COVID-19 had the highest acceptance rate of the MPOX vaccine compared to 81.0% of those who rejected the COVID-19 vaccine, *p* < 0.001. Previous MPOX is associated with higher vaccine acceptance compared to those who did not contract infection (61.1% vs. 38.9%, *p* = 0.03%) (Table 2).

The 5C scores of the psychological antecedents between those who reject and those who accept the MPOX vaccine, were detected as follows: confidence ((5.3 ± 1.3) versus (6.2 ± 0.7), *p* < 0.001); constraints ((4.4 ± 1.5) versus (4.2 ± 1.7), *p* = 0.059); complacency ((4.0 ±1.7) versus (3.8 ± 1.9), *p* = 0.091); calculation ((5.7 ± 1.1) versus (5.9 ± 1.1), *p* = 0.015); and collective responsibility ((55.1 ± 1.0) versus (5.6 ± 0.9), *p* < 0.001) (Table 2).

### 3.4. The Psychological Antecedents of HCWs and the General Population toward MPOX Vaccine

The 5C scores of psychological antecedents between HCWs and the general population, respectively, were detected as follows: confidence ((5.9 ± 1.1) versus (5.6 ± 1.9), *p* = 0.022); constraints ((4.2 ± 1.7) versus (4.4 ± 1.6), *p* = 0.117); complacency ((3.8 ± 1.9) versus (3.9 ± 1.7), *p* = 0.545); calculation ((5.9 ± 1.1) versus (5.7 ± 1.1), *p* = 0.156); and collective responsibility ((5.5 ± 0.9) versus (5.2 ± 1.0), *p* = 0.001). (Figure 1).

### 3.5. Determinants of MPOX Vaccine Acceptance among the Studied Ghanaian Population

Age was a significant determinant of VH in the unadjusted odds ratio (OR =1.03 (95% CI, 1.01–1.06, *p* = 0.013), while in the adjusted model, it became not significant (AOR = 1.02, 95% CI 0.99–1.05, *p* = 0.123). Male gender was associated significantly with increased vaccine acceptance by 48% compared to females (AOR = 1.48, 95% CI, 1.00–2.18, *p* = 0.049). Being non-HCWs was significantly associated with increased VH (AOR = 0.58, 95% CI, 0.41–0.81, *p* = 0.002), while in the regression model, this was not statistically significant (AOR = 0.86, 95% CI, 0.57–1.29, *p* = 0.468). Urban residence increased the adjusted odds of MPOX vaccine acceptance (AOR = 0.63, 95% CI, 0.41–0.96, *p* = 0.033). Refusal of the COVID-19 vaccine significantly reduced acceptance of the MPOX vaccine (AOR = 0.29, 95% CI, 0.16–0.52, *p* < 0.001). Confidence in vaccination and collective responsibility were significantly associated with MPOX vaccine acceptance ((AOR = 2.45, 95% CI, 1.93–3.15, *p* < 0.001) and (AOR = 1.34, 95% CI, 1.02–1.75, *p* = 0.034)), respectively (Table 3).

## 4. Discussion

For HCWs who are at high risk of MPOX exposure, vaccination prior to exposure is advised [28] even though the WHO does not consider mass MPOX vaccination to be a necessary step [28]. The ongoing spread of numerous infectious diseases, including MPOX, is well known to be a major problem, and increasing vaccination rates is widely acknowledged as a key strategy [29,30].

In this study, we aimed to determine the intention of the Ghanaian population to accept the MPOX vaccine, including the HCWs. Through the snowball sampling method, we recruited 605 participants, and around one-third of them was comprised of HCWs. The overall intention to receive the MPOX vaccine among the studied population was 46.1%. The acceptance rate was higher among HCWs compared to the general population. We reported significantly higher scores of confidence and collective responsibility among HCWs compared to the general population. The main identified predictors of vaccine acceptance were male gender, living in an urban area, attitude toward COVID-19 vaccination, confidence in MPOX, and collective reasonability.

Due to their frontline position in caring for infected patients, HCWs are at risk of MPOX acquisition [31]. Therefore, evaluating HCWs’ attitudes toward MPOX vaccination is a crucial step in efforts to stop the increase in MPOX cases [32]. In addition to the critical role that HCWs play in outbreak response and community education, their attitudes toward vaccination can affect the vaccine recommendations that they make to patients. Regarding MPOX vaccinations, HCWs in the Czech Republic revealed poor levels of factual knowledge. Furthermore, there were various misconceptions among the participants on topics such as the availability of effective MPOX vaccines and antivirals, the risk of vertical transmission, and homosexual stigma. The most important characteristics for predicting MPOX vaccination uptake were cues to action and perceived susceptibility [33]. In this study, we found that the overall acceptance of the MPOX vaccine among the general population was relatively low (41.7%). This may be reflected in the slightly higher acceptance among HCWs (55.3%). Curiously, despite the widespread belief that healthcare professionals, particularly physicians, should accept it completely, this study found that around one-half of HCWs were accepting of vaccination. This finding is in line with a recently published meta-analysis that assessed VH among 8045 participants. The overall acceptance rate for the monkeypox vaccination was 56.0% (95% CI: 42.0–70.0%). In the general population, the prevalence of vaccine acceptance was 43.0% (95% CI: 35.0–50.0%), among healthcare professionals it was 63.0% (95% CI: 42.0–70.0%), and among lesbian, gay, bisexual, and transgender (LGBTI) people, it was 84.0% (95% CI: 83.0–86.0%) [34]. In fact, people’s perceptions regarding vaccination acceptance may be influenced by a variety of local, racial, religious, cultural, and other factors, as well as false information, as was evidently seen during the COVID-19 pandemic [14,15].

### 4.1. Confidence 

One of the main identified determinants of vaccine acceptance was confidence in the vaccine and healthcare system. In this study, vaccine confidence increased the odds of MPOX vaccine acceptance (OR = 1.93–3.15, *p* < 0.001). Nevertheless, there are a number of variables that are linked to vaccine confidence, including many others that are of great concern, such as the safety and efficacy of the vaccine [35]. So, we recommend that effectiveness research is urgently needed to better understand the true impact of the monkeypox vaccines as the outbreak develops. A recent study in the Netherlands discovered that among those who had not received a booster vaccination, a two-shot immunization series with the modified vaccinia virus Ankara-Bavarian Nordic (MVA-BN, also known as Jynneos, Imvanex, or Imvamune) estimated to partially low titers of MPOXV-protective neutralizing antibodies. A third dose of the same vaccine significantly improves the immune response to the antibodies, whereas dose-sparing the MVA-based influenza vaccine results in low levels of MPOXV-neutralizing antibodies. Moreover, it is important to evaluate vaccine protection in populations at risk. The significance of MPOXV-neutralizing antibodies as a potential protection correlating with disease and transmission has not been yet to be precisely defined [36].

### 4.2. Collective Reasonability

Collective responsibility was positively correlated with vaccination intentions. In this study, we found that collective reasonability was a significant determinant of MPOX vaccine acceptance. Similarly, Ulloque-Badaracco et al. [34] reported that participants with high collective responsibility reported a significantly greater uptake of the COVID-19 vaccine (64.7 vs. 49.7%). Further evidence was proven in Hong Kong and the United Kingdom (UK). A study in Hong Kong discovered that nurses with higher collective reasonability scores had higher intentions to vaccinate against COVID-19 and to take the influenza vaccine [37]. A study in the UK discovered that older adults with lower collective reasonability scores were less likely to take the shingles, pneumococcal, and influenza vaccines [38]. Finally, among Algerians, it was clear that those who cared about their families, friends, and patients were more motivated to take booster doses. Expert recommendations (24.6%) and the belief that the COVID-19 vaccine boosters were necessary and efficient were the most common reasons for the acceptance of COVID-19 vaccination, while rejection was primarily due to the belief that primer doses are sufficient (15.5%), or that vaccination in general is inefficient (8%) [39].

### 4.3. Refusal of Other Vaccines

We found that participants who refused to receive the COVID-19 vaccine were refusing the MPOX vaccine. Surprisingly, there does not appear to be any connection between attitudes toward MPOX vaccination and socioeconomic status or level of education. We speculated that high levels of conspiracy-theory thinking, a low threshold for interference with one’s sense of personal freedom, aversion to needles or blood, and religious concerns are much better predictors of such a rejection of both vaccines. In addition, we think that worried participants are the most frequent recipients of misinformation. Social media have been cited as a potential source of misinformation by experts. Over the last decade, there has been phenomenal growth in worldwide Internet access, unparalleled development of social media platforms, and the opportunity to freely share news, films, and ideas online. This allows anyone with a smartphone or other device linked to the Internet to voice their opinion on certain diseases, such as MPOX or COVID-19 [9]. The importance of education and the promotion of vaccines is highlighted by all these findings. Not only the general public should be informed about the effectiveness and safety of available vaccines against monkeypox, but healthcare professionals, in particular, should also be informed because they are at a higher risk. Because opinions about various vaccines, as well as misinformation and trust, are likely to change, they must be continually assessed in light of this fact.

### 4.4. Strengths and Limitations

To the best of our knowledge, this is the first study to evaluate the psychological influencing factors for the MPOX vaccine among the Ghanaian population, including HCWs. First, the internal consistency and reliability of the study’s findings were increased by the use of the validated 5C scale questionnaire. The non-random sampling technique, one of the study’s many limitations, may make it difficult to generalize the study’s findings. Second, the cross-sectional survey has its own inherent limitations. The respondents are vulnerable to reporting bias; we included only those who have access to the Internet or have smartphones. In addition, we did not include the illiterate population who represents around one-fifth of the population. However, we adopted such a sampling technique for the sake of feasibility. We did not ask about the human immunodeficiency virus (HIV) infection. In fact, an associated HIV infection may affect the responses of the studied population as those who are infected are more susceptible to the disease. We were unable to address causality, and the results of this survey only reflect a single time point that may change over time. In addition, some questions were very subjective, such as about the income level. Finally, the study’s small sample size should be considered another drawback. However, the study can serve as a starting point and inspiration for further research that aims to thoroughly analyze the problem of MPOX vaccination and its effective application, particularly in areas where the virus caused outbreaks. In fact, identifying particular groups with lower rates of intention to vaccinate against monkeypox would aid governments and health authorities in exploring and developing more effective public health approaches for vaccination. The findings of our study provide information for the global monkeypox vaccination program. They could serve as a blueprint for creating new public health regulations targeted at a population with low acceptance rates for the monkeypox vaccine. Additionally, these findings could serve as a guide for stratifying populations with low vaccine acceptance and developing targeted strategies for them in outbreaks to come.

## 5. Conclusions

Monkeypox as a public health of concern has witnessed an increase in the number of cases with confirmed fatalities during the ongoing 2022 outbreak. The participants in this study did not show high levels of intention to accept the MPOX vaccination. This negative attitude was more observed among the general population compared to HCWs. Different sociodemographic characteristics positively affected vaccine acceptance, such as the male sex and rural resident. Receiving the COVID-19 vaccine was positively associated with MPOX vaccine acceptance. Moreover, confidence in the vaccine and collective responsibility was the main psychological antecedents that affected participants’ intention to receive the vaccination. Consequently, tailoring the efforts aiming to promote MPOX vaccination is needed especially among non-HCWs through upscaling their confidence in vaccine effectiveness and safety and promoting their attitude about the importance of self-vaccination to protect others.

## Figures and Tables

**Figure 1 vaccines-11-00240-f001:**
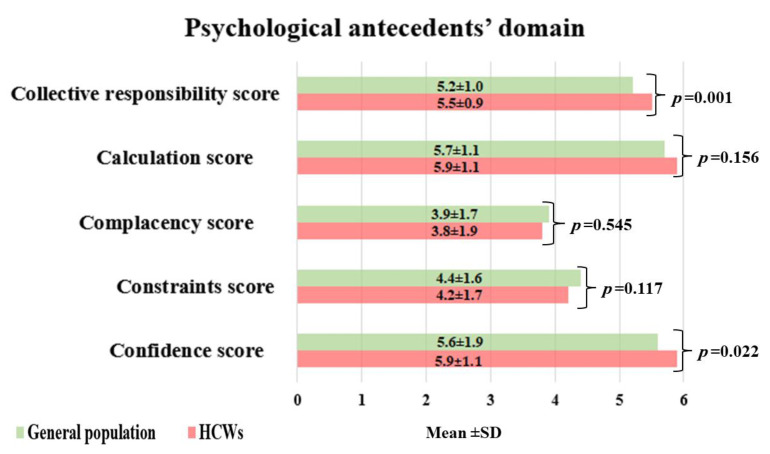
The psychological antecedents of the general population versus HCWs.

**Table 1 vaccines-11-00240-t001:** Sociodemographic criteria of the studied population of Ghana.

Variables		Overall (N = 605)
Age	Mean ± SD	30.0 ± 6.8
	[Min, Max]	[18.0, 67.0]
Sex	Female	237 (39.2)
	Male	368 (60.8)
Residence	Urban	152 (25.2)
	Rural	450 (74.8)
Marital status	Have a partner	40 (6.6)
	Married	153 (25.3)
	Single	412 (68.1)
Education	Pre-college/high school	82 (13.6)
	Professional/technical	20 (3.3)
	Bachelor degree	290 (47.9)
	Diploma	161 (26.6)
	Post-graduate	52 (8.6)
Income	Low income	327 (54.0)
	Middle income	268 (44.3)
	Upper income	10 (1.7)
Region	Greater Accra	314 (51.9)
	Ashanti	113 (18.7)
	Other	178 (29.4)
Occupation	Not healthcare worker	408 (67.4)
	Nurse/midwife	52 (8.6)
	Pharmacy technician	7 (1.2)
	Physician/physician assistant	27 (4.5)
	Public health personal	103 (17.0)
	Laboratory technician	8 (1.3)
Chronic diseases	No	578 (95.5)
	Yes	27 (4.5)
COVID-19 vaccination	Fully vaccinated	399 (66.0)
	Not going to take the vaccine	105 (17.3)
	Took the first dose, going to take the second	84 (13.9)
	Took the first dose and will not take the second	17 (2.8)
Know someone passed due to MOPX	I do not know	71 (11.8)
	No	518 (85.6)
	Yes	16 (2.6)
MPOX infection	No	551 (91.1)
	Yes	54 (8.9)
Psychological antecedents	Confidence score	5.7 ± 1.1
	Constraints score	4.3 ± 1.6
	Complacency score	3.9 ± 1.8
	Calculation score	5.8 ± 1.1
	Collective responsibility score	5.3 ± 1.0

**Table 2 vaccines-11-00240-t002:** Different sociodemographic criteria and participants’ attitudes across their intention to receive a vaccination, (n = 605).

Demographic Characteristics Total (N = 605)		Reject Vaccination N (%)	Accept Vaccination N (%)	*p*-Value
Age	Mean ± SD	29.4 ± 6.5	30.8 ± 7.1	0.013
Sex	Female	143 (60.3)	94 (39.7)	0.013
	Male	183 (49.7)	185 (60.3)	
Residence	Rural	72 (47.4)	80 (52.6)	0.078
	Urban	254 (56.4)	199 (43.8)	
Marital status	Have a partner	23 (57.5)	17 (42.5)	0.278
	Married	74 (48.4)	79 (51.6)	
	Single	229 (55.6)	183 (44.4	
Educational level	Pre-college/high school	42(51.2)	40(48.8)	0.264
	Professional/technical	11(55.0)	9(45.0)	
	Undergraduate (Bachelor)	165(56.9)	125(43.1)	
	Diploma	77(47.8)	84(52.2)	
	Post-graduate	31(50.6)	21(40.4)	
Income	Low income	168 (51.4)	159 (48.6)	0.369
	Middle income	153 (57.1)	115 (42.9)	
	Upper income	5 (50.0)	5 (50.0)	
Region	Ashanti	62 (54.9)	51 (45.1)	0.196
	Greater Accra	178 (56.7)	136 (43.3)	
	Others	86 (48.3)	92 (51.7)	
Occupation	HCWs	88 (44.7)	109 (55.3)	<0.001
	Not healthcare worker	238 (58.3)	170 (41.7)	
Chronic diseases	No	314 (54.3)	264 (45.7)	0.418
	Yes	12 (44.4)	15 (55.6)	
COVID-19 vaccination	Fully vaccinated	182 (45.6)	217 (54.4)	<0.001
	Not going to take the vaccine	85 (81.0)	20 (19.0)	
	Took first dose, going to take the second	47 (56.0)	37 (44.0)	
	Took first dose, will not take the second	12 (70.6)	5 (29.4)	
Know someone who passed due to MPOX	I do not know	41 (57.7)	30 (42.3)	0.155
	No	280 (54.1)	238 (45.9)	
	Yes	5 (31.3)	11 (68.8)	
MPOX infection	No	305 (55.4)	246 (44.6)	0.030
	Yes	21 (38.9)	33 (61.1)	
Psychological antecedents	Confidence score	5.3 ± 1.3	6.2 ± 0.7	<0.001
	Constraints score	4.4 ± 1.5	4.2 ± 1.7	0.059
	Complacency score	4.0 ± 1.7	3.8 ± 1.9	0.091
	Calculation score	5.7 ± 1.1	5.9 ± 1.1	0.015
	Collective responsibility score	5.1 ± 1.0	5.6 ± 0.9	<0.001

**Table 3 vaccines-11-00240-t003:** Multivariate analysis of the determinant of MPOX vaccine hesitancy.

Independent Variables				Unadjusted OR	Adjusted OR
Age	Mean (SD)	29.4 (6.5)	30.8 (7.1)	1.03 (1.01–1.06, *p* = 0.013)	1.02 (0.99–1.05, *p* = 0.123)
Gender	Female	143 (60.3)	94 (39.7)	-	-
	Male	183 (49.7)	185 (50.3)	1.54 (1.11–2.14, *p* = 0.011)	1.48 (1.00–2.18, *p* = 0.049)
Occupation	HCWs	88 (44.7)	109 (55.3)	-	-
	Not healthcare worker	238 (58.3)	170 (41.7)	0.58 (0.41–0.81, *p* = 0.002)	0.86 (0.57–1.29, *p* = 0.468)
Residence	Rural	72 (47.4)	80 (52.6)	-	-
	Urban	254 (56.1)	199 (43.9)	0.71 (0.49–1.02, *p* = 0.063)	0.63 (0.41–0.96, *p* = 0.033)
COVID-19 vaccination	Fully vaccinated	182 (45.6)	217 (54.4)	-	-
	Not going to take the vaccine	85 (81.0)	20 (19.0)	0.20 (0.11–0.33, *p* < 0.001)	0.29 (0.16–0.52, *p* < 0.001)
	Took first dose, going to take the second	47 (56.0)	37 (44.0)	0.66 (0.41–1.06, *p* = 0.086)	0.77 (0.45–1.31, *p* = 0.331)
	Took first dose, will not take the second	12 (70.6)	5 (29.4)	0.35 (0.11–0.96, *p* = 0.052)	0.65 (0.18–2.09, *p* = 0.480)
MPOX infection	No	305 (55.4)	246 (44.6)	-	-
	Yes	21 (38.9)	33 (61.1)	1.95 (1.11–3.50, *p* = 0.022)	1.88 (0.99–3.63, *p* = 0.056)
Confidence score	Mean (SD)	5.3 (1.3)	6.2 (0.7)	2.65 (2.14–3.33, *p* < 0.001)	2.45 (1.93–3.15, *p* < 0.001)
Constraints score	Mean (SD)	4.4 (1.5)	4.2 (1.7)	0.91 (0.82–1.00, *p* = 0.059)	0.84 (0.69–1.03, *p* = 0.088)
Complacency score	Mean (SD)	4.0 (1.7)	3.8 (1.9)	0.93 (0.85–1.01, *p* = 0.091)	1.07 (0.88–1.29, *p* = 0.513)
Calculation score	Mean (SD)	5.7 (1.1)	5.9 (1.1)	1.20 (1.04–1.39, *p* = 0.016)	0.93 (0.75–1.14, *p* = 0.482)
Collective responsibility score	Mean (SD)	5.1 (1.0)	5.6 (0.9)	1.85 (1.55–2.23, *p* < 0.001)	1.34 (1.02–1.75, *p* = 0.034)

## Data Availability

All data are available upon request by emailing the first author.

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
