# Peer review of "Monkeypox Vaccine Acceptance among Ghanaians: A Call for Action"

_vaccines, 2023, doi:10.3390/vaccines11020240_

Round 1

Reviewer 1 Report

Dear Author(s),

Thank you so much for this timely and interesting work that aim to increase our collective knowledge about mpox vaccine acceptance determinants in Ghana as a representative for endemic countries.

1. Please use the new name of monkeypox (mpox) in the entire manuscript.

2. The manuscript requires language editing and proofreading. I found several style and grammar issues.

3. Line 28: remove repetition.

4. Line 96: I am unaware of using the 5C model in several studies on mpox. The only study that I have read recently was this one:
https://www.mdpi.com/2076-393X/10/12/2151
Please cite it at the end of the sentence.

5. The study's overall aim (goal) and primary and secondary objectives should be explicitly stated at the end of the Introduction section.

6. Please follow the STROBE guidelines in structuring your Methods section.

7. Please cite the STROBE guidelines accordingly within your Methods section.
Suggested ref (optional):
https://doi.org/10.1016/S0140-6736(07)61602-X

8. Please upload the STROBE checklist for cross-sectional studies as a supplementary file:
https://www.strobe-statement.org/checklists/

9. Line 117 - 119: what was the prevalence rate (P) that you assumed in this study?

10. Line 117 - 119: why the design effect was 1.5?

11. What are the psychometric properties of the data collection tool (questionnaire) that you used in this study?

12. For transparency, please upload the questionnaire, if possible, as a supplementary file.

13. Line 165: the study was conducted among the Ghanaian population living in Ghana; why was ethical clearance obtained from another country? Is there any explanation for that?

14. Table 1 requires a bit of organization, it is not easy to read.

15. According to the World Bank (WB), Ghana's literacy level in 2018 was 79% which means that you have inadvertently excluded 21% of the Ghanaian population. This should be clearly stated as a limitation of this study. I guess that you may have a good justification for this limitation which is methodological feasibility, as it is not easy or cheap to survey people without functional literacy.

16. Can you make a map chart for the respondents' residences?

17. In the main text, there is no need to have % + (no of subjects). It is enough to mention % only.
For example, in Line 181: (About 53.9% of the studied Ghanian population).

18. Figure 1 has a poor quality and it is not really useful. Remove it better.

19. Line 189: Males were more intended? Please revise the language of the manuscript entirely.

20. The Discussion was well written; however, it may benefit from reflecting on similar studies on mpox.
Suggested refs (optional):
https://doi.org/10.3390/vaccines10122022

21. Line 266-269: I agree that the role of the infodemic and misinformation on vaccine confidence is decisive. This point may need further elaboration.

22. Line 287: collective responsibility. In the studies of healthcare workers' attitudes towards COVID-19 booster vaccines, it was clear that those who care about their families, friends, and patients were more motivated to take booster doses. You can strengthen your narrative by reflecting on this point too.
Suggested ref (optional):
https://doi.org/10.3390/vaccines10040621

Sincerely,

Author Response

Dear estimated Editor,

Regarding the manuscript with ID: Vaccines, entitled “Monkeypox vaccine acceptance among Ghanaian; a call for action”.

We are deeply thankful and grateful for the comprehensive and constructive review reports that were highly insightful and enabled the improvement of the quality of the manuscript. In the following pages are our point-by-point responses to the comments raised by the reviewers.

Please find attached a revised version of the manuscript ““Monkeypox vaccine acceptance among Ghanaian; a call for action”. The revisions were highlighted using the "Highlight Changes" function in the manuscript file.

I hope that the revisions in the manuscript and the accompanying responses will be sufficient to make the manuscript suitable for publication in Vaccines

Sincerely, and on behalf of co-authors

ramy_ghazy@alexu.edu.eg

Reviewer 1

Dear reviewer we would like to thank you for these fruitful comments that have greatly improved the quality of our work. Regarding your comment on the statistical analysis, we revised all the result section and the whole manuscript to match the good reputation of the journal.

  1. Please use the new name of monkeypox (mpox) in the entire manuscript.

Response: Thank Prof Abanob, we have corrected this abbreviation.

  1. The manuscript requires language editing and proofreading. I found several style and grammar issues.

Response: Sorry for inconvenience, we have gone through the manuscript and corrected these typos.

  1. Line 28: remove repetition.

Response: Corrected

  1. Line 96: I am unaware of using the 5C model in several studies on mpox. The only study that I have read recently was this one:
    https://www.mdpi.com/2076-393X/10/12/2151
    Please cite it at the end of the sentence.

Response: Thank you prof, indeed, this work is done by our team, when we submitted the current work, it was not published yet. We have already cited this work based on your recommendation.

  1. The study's overall aim (goal) and primary and secondary objectives should be explicitly stated at the end of the Introduction section.

Response: We agree with you. Correction made.

  1. Please follow the STROBE guidelines in structuring your Methods section.

Response: Done.

  1. Please cite the STROBE guidelines accordingly within your Methods section.
    Suggested ref (optional):
    https://doi.org/10.1016/S0140-6736(07)61602-X

Response: added and cited.

  1. Please upload the STROBE checklist for cross-sectional studies as a supplementary file:
    https://www.strobe-statement.org/checklists/

Response: Done and uploaded as supplementary material.

  1. Line 117 - 119: what was the prevalence rate (P) that you assumed in this study?

 Response:  as there was any published study before we supposed it 550 to gain the largest sample size.

  1. Line 117 - 119: why the design effect was 1.5?

Response: Sorry for this mistake, it should be 1 as we did not use stratified or cluster sampling technique. We corrected this mistake.

  1. What are the psychometric properties of the data collection tool (questionnaire) that you used in this study?

Response: The questionnaire was proved to have internal consistency as all domains had Cronbach’s alpha > 0.7.  Moreover, we tested the reliability of the tool in this study, it exceeded 0.8 for all items.

  1. For transparency, please upload the questionnaire, if possible, as a supplementary file.

Response: Done and uploaded as supplementary material.

  1. Line 165: the study was conducted among the Ghanaian population living in Ghana; why was ethical clearance obtained from another country? Is there any explanation for that?

Response: This study was part of large project to assess vaccine hesitancy toward MOPX in African countries, as the principal investigator was from Egypt, he got ethical approval from there covering all African countries.

  1. Table 1 requires a bit of organization; it is not easy to read.

Response: Sorry for the inconvenience. Rearranged to be more compensable.

  1. According to the World Bank (WB), Ghana's literacy level in 2018 was 79% which means that you have inadvertently excluded 21% of the Ghanaian population. This should be clearly stated as a limitation of this study. I guess that you may have a good justification for this limitation which is methodological feasibility, as it is not easy or cheap to survey people without functional literacy.

Response: We agree with you and add this point to the limitation.

  1. Can you make a map chart for the respondents' residences?

Response:  We appreciate these comments but we think this figure may be of no value as there are 16 regions, and most of the participants were from the 2 main largest cities of Ghana (Ashanti, and Greater Accra).

Reg

Frequency

Percent

Valid Percent

Cumulative Percent

Valid

AHAFO

5

.8

.8

.8

Ashanti

113

18.7

18.7

19.5

BONO EAST

9

1.5

1.5

21.0

BRONG AHAFO

7

1.2

1.2

22.1

CENTRAL

36

6.0

6.0

28.1

EASTERN

37

6.1

6.1

34.2

Greater Accra

314

51.9

51.9

86.1

NORTH EAST

4

.7

.7

86.8

NORTHERN

13

2.1

2.1

88.9

OTI

9

1.5

1.5

90.4

SAVANNAH

7

1.2

1.2

91.6

UPPER EAST

7

1.2

1.2

92.7

UPPER WEST

5

.8

.8

93.6

Volta

22

3.6

3.6

97.2

WESTERN

13

2.1

2.1

99.3

WESTERN NORTH

4

.7

.7

100.0

Total

605

100.0

100.0

  1. In the main text, there is no need to have % + (no of subjects). It is enough to mention % only.
    For example, in Line 181: (About 53.9% of the studied Ghanian population).

Response: Corrected

  1. Figure 1 has poor quality and is not really useful. Remove it better.

Response: Removed

  1. Line 189: Males were more intended? Please revise the language of the manuscript entirely.

Response: Corrected

  1. The Discussion was well written; however, it may benefit from reflecting on similar studies on mpox.
    Suggested refs (optional):
    https://doi.org/10.3390/vaccines10122022

Response: Done. Reference added.

Line 266-269: I agree that the role of the infodemic and misinformation on vaccine confidence is decisive. This point may need further elaboration.
Response: Done

  1. Line 287: collective responsibility. In the studies of healthcare workers' attitudes towards COVID-19 booster vaccines, it was clear that those who care about their families, friends, and patients were more motivated to take booster doses. You can strengthen your narrative by reflecting on this point too.
    Suggested ref (optional):
    https://doi.org/10.3390/vaccines10040621

Response: Done. Reference added.

Reviewer 2 Report

Accept after fine/minor english language spell check revision.

Author Response

  1. Accept after fine/minor English language spell check revision.

Response: Thanks Doctor. We went through the manuscript and corrected the lingual and grammatic mistakes.

Reviewer 3 Report

This study is a cross-sectional online survey conducted in Ghana via social media platforms such as WhatsApp, LinkedIn, Telegram, and Facebook. The authors analyzed demographical social and clinical factors associated with the acceptance of MPX vaccine. This is an interesting study performed in an African country where MPX was largely spreading.

Although the methods are quite clear, I suggest to clarify some points.

At line 21, Monkeypox (MPX) should be mentioned for the first time. “Ghana ranked the 31st worldwide, and 3rd in Africa in the number of confirmed Monkeypox (MPX) cases”.

At lines 115-117 the formula is not clear. Please provide a better explanation of this formula and its use in clinical practice. Provide some references in this regard.

At lines 121 “A self-administered questionnaire, in English language, with two main sections that

is tested and validated beforehand was used in this study”: you must specify when this questionnaire was used (for istance the previous studies that performed this tool)

Since MPX infection often affects young males who often have HIV infection, the authors should clarify whether they have evaluated the degree of vaccine acceptance of MPX in this setting. Otherwise, this should be added among the limitations.

Author Response

Reviewer 3

This study is a cross-sectional online survey conducted in Ghana via social media platforms such as WhatsApp, LinkedIn, Telegram, and Facebook. The authors analyzed demographical social and clinical factors associated with the acceptance of MPX vaccine. This is an interesting study performed in an African country where MPX was largely spreading.

Response: Thank you for your time and effort. We appreciate your fruitful suggestions and all of them were considered.

  1. Although the methods are quite clear, I suggest to clarify some points. At line 21, Monkeypox (MPX) should be mentioned for the first time. “Ghana ranked the 31st worldwide, and 3rd in Africa in the number of confirmed Monkeypox (MPX) cases”.

Response: Done

  1. At lines 115-117 the formula is not clear. Please provide a better explanation of this formula and its use in clinical practice. Provide some references in this regard.

Response: We agree with you. The equation was added and cited.

  1. At lines 121 “A self-administered questionnaire, in English language, with two main sections that is tested and validated beforehand was used in this study”: you must specify when this questionnaire was used (for istance the previous studies that performed this tool).

Response: The 5C scale was used in many studies to assess vaccine hesitancy as we mentioned in the introduction. Based on your recommendation.

  1. Since MPX infection often affects young males who often have HIV infection, the authors should clarify whether they have evaluated the degree of vaccine acceptance of MPX in this setting. Otherwise, this should be added among the limitations.

Response: Sorry for inconvenience. This point was added to the limitation.

Round 2

Reviewer 1 Report

Dear authors,

Thank you for addressing all my previous comments appropriately. The manuscript is in a better format now.

Sincerely,